# Combinatorial Therapy: Targeting CD133+ Glioma Stem-like Cells with a Polysaccharide–Prodrug Complex Functionalised Gold Nanocages

**DOI:** 10.3390/biomedicines12050934

**Published:** 2024-04-23

**Authors:** Sreejith Raveendran, Amit Giram, Mehrnaz Elmi, Santanu Ray, Christopher Ireson, Mo Alavijeh, Irina N. Savina

**Affiliations:** 1School of Pharmacy and Biomolecular Sciences, University of Brighton, Moulsecoomb, Lewes Road, Moulsecoomb, Brighton BN2 4GJ, UK; 2Pharmidex Pharmaceutical Services Limited, 167-169 Great Portland Street, Fifth Floor, London W1W 5PF, UK; 3School of Environmental Sciences, University of Brighton, Moulsecoomb, Lewes Road, Moulsecoomb, Brighton BN2 4GJ, UK

**Keywords:** glioma, glioblastoma, prodrug, polysaccharide, combinatorial therapy, cancer therapy, targeted drug delivery

## Abstract

Cancer treatments are advancing to harness the body’s immune system against tumours, aiming for lasting effects. This progress involves combining potent chemotherapy drugs with immunogens to kill cancer cells and trigger lasting immunity. Developing new prodrugs that integrate both chemotherapy and immune-boosting elements could significantly improve anticancer outcomes by activating multiple mechanisms to kill cancer cells. While bacterial polysaccharides are typically not used in therapy due to their immune-stimulating properties, we propose a safe application of an extremophilic bacterial polysaccharide, Mauran (MR), modified with the anticancer drug 5-fluorouracil (5FU) to create a novel prodrug. This obtained prodrug, chloracetyl-MR-5FU, is specifically targeted using gold nanocages to CD133+ glioma cells. Test results have shown a high encapsulation efficiency of the drug during the polysaccharide modification process; its anticancer activity was demonstrated in vitro and the release of the prodrug was demonstrated in ex vivo studies.

## 1. Introduction

New combination tactics are replacing single-drug therapy for cancer since nanotechnology began to impact pharmaceutics and biotherapeutics in recent decades. [1]. There have been several novel targeting strategies introduced recently along with chemo in combination with immuno-, cell and gene therapies [2]. This has greatly increased the chance of achieving the target and bringing more effective therapeutic benefits and a more sustained immunological memory. Nanocarriers not only play an important role in delivering the cargo of our choice but sometimes they interfere with the cellular dynamics as well [3,4]. Carriers can be selected based on the purpose, chemistry, mechanism and selectivity of the payload [5]; moreover, the size and shape of the nanoparticles are also important if our target is across the blood-brain barrier. Gold nanoparticles, being accepted as universal nano cargo delivery agents that have exceptionally good physicochemical properties [6,7,8] like surface plasmon resonance, as well as good tunability in morphological parameters, are used as the nanocarrier in this study. Gold nanocages (AuNcgs) were shown to be an idealistic carrier not only for drug delivery but for several diagnostic imaging fields [9] and hyperthermia therapies [10]. Moreover, the hollow nanocage from the Au–Ag alloy complex is well-studied for its myriad of biomedical and nanodrug delivery applications [3,9,11,12,13].

Glioblastoma (GBM) is one of the major malignant brain tumours and so far, patients with GBM and brain metastasis have extremely less treatment options [14,15,16,17,18]. Existing therapies have many difficulties in the complete killing of GBM cells as they possess a mixed grade of cells. Certain cells within these highly malignant tumours may respond to primary therapies while others do not. GBMs are aggressive with a near-sure recurrence despite aggressive surgical resection and combinatorial chemoradiation [19]. Increasing evidence suggests that these tumours harbour a brain tumour-initiating cell (BTIC) population that is resistant to radiation and current standard chemotherapies [20,21,22]. Interestingly, this highly malignant population can be identified using the CD133 (Prominin-1) cell surface marker [14,23,24,25]. These markers are well studied for their existence and contribution of stemness to several types of BTICs and other types of cancers like osteosarcoma and colon cancers [23,26]. Essentially, these CD133 antigens play a crucial role in the heterogenicity of tumours, recurrence and chemoresistance [24,27,28]. CD133+ cancer stem cells (CSC) were identified and targeted using several nano delivery methods [29,30] like aptamers [26], and monoclonal antibodies [31]. Based on the highest probability of targeting all possible BTICs of GBM, we chose CD133 as our potential target for this study.

The most widely used pyrimidine pharmacological derivative of fluorouracil is 5-fluorouracil (5FU), a well-known broad-spectrum anticancer chemotherapeutic agent used to treat pancreatic, oesophageal, stomach, skin, and intestinal malignancies. [32,33]. 5FU is a drug that has low bioavailability and high side effects, which on nanoencapsulation can be altered to a greater extent [34]. Moreover, it is a highly cost-effective anticancer drug currently being studied for its potential nanodrug delivery application in brain tumour treatment [35,36]. To increase the selectivity and therapeutic benefits as well as decrease the toxicity, 5FU was used to design a novel polysaccharide prodrug. This was achieved via the chemical modification of a highly sulphated extremophilic bacterial polysaccharide called Mauran (MR) and the functionalisation of 5FU to it [34,37,38,39]. MR was extracted from a moderately halophilic bacterium, *Halomonas maura*, and roughly purified before its chemical modification [34]. These MR heteropolymers are chemically, physically and biologically versatile polysaccharides, constituting monosaccharide residues bound via glycosidic linkages. They have been explored for several bioactive properties in the last two decades [39]. Our previous research works have demonstrated MR as an active surface-enhancing biopolymer for nanoparticle passivation and targeted drug delivery to cancers [40]. MR was acclaimed as an immunomodulating polysaccharide in the mid 1960s; however, further immunological evaluation is required [39]. In addition to the 5FU selectivity, another major objective of this sugar modification is to establish the therapeutic benefit of MR on galectin-3 (Gal-3) expressing cancer cells as a prodrug of 5FU. Gal-3 is a conserved carbohydrate recognition domain that binds ß-galactosidase. It regulates several cellular functions like proliferation, apoptosis, migration, angiogenesis, tumorigenesis, etc. [41]. Carbohydrate prodrugs are considered to have therapeutic benefits on Gal-3 expressing cancers like breast, colon, lung, bladder, prostate, thyroid, pancreas, lymphoma, head and neck and other gastrointestinal cancers [37]. Here, we test the function of a similar carbohydrate prodrug developed against glioma, GL261.

We have chosen microwave-synthesised AuNcgs [42,43] in the central core for the design of Therapeutic Mauran–Fluorouracil Gold (TMFG) nanocages. Briefly, here we present the synthesis of a novel polysaccharide prodrug, MR-5FU functionalised AuNcg formulation, with a targeting monoclonal antibody, an anti-CD133 antibody, for the treatment of glioblastoma. The steps involved in the synthesis were briefed in the schematic representation (Figure 1). While targeting the BTICs with the CD133 antibody for the site-specific delivery of nanocage cargo, holding an extremophilic sulphated polysaccharide (SPS)-based novel prodrug complex is reported for the first time in the paper as a novel strategy in cancer combinatorial therapy.

## 2. Materials and Methods

### 2.1. Materials

Gold (III) chloride, silver nitrate, ethylene glycol, 5-fluorouracil, sodium sulphide nonahydrate and polyvinyl pyrrolidone were purchased from Sigma-Aldrich (St. Louis, MO, USA). All other reagents used were of analytical grade. GL261, mouse glioblastoma cell line (#ACC802) was purchased from DSMZ, Braunschweig, Germany. Anti-CD133 antibody was purchased from Fisher Scientific (#11-1331-82, Madison, WI, USA). Dulbecco’s modified Eagle’s medium (DMEM) and foetal bovine serum (FBS) was purchased from Sigma-Aldrich and Gibco (Waltham, MA, USA), respectively. CellTiter 96Aqueous- one solution (#G3582), Promega Corporation (Madison, WI, USA), was purchased for the MTS assay. The extremophilic polysaccharide Mauran, used for the study, was provided from Toyo University, Toyo, Japan.

### 2.2. Synthesis and Characterisation of Hollow AuNcgs

AuNcgs were synthesised using the galvanic replacement reaction of silver nanocubes (AgNcbs) with Au, using a microwave-assisted heating method as mentioned elsewhere [43]. In brief, AgNcbs were synthesised with silver nitrate and polyvinylpyrrolidone (PVP) using microwave heating via the start and stop method. After several washing steps with an ethanol/water mixture (1:1, *v*/*v*), followed by water, the AgNcbs harvested were dispersed in water before the galvanic replacement reaction. A total of 0.5 mL of AgNcb solution was dispersed in 5 mL of water and added with 2.5 mL of 1 mM HAuCl_4_ solution after preheating in a microwave oven at 800 W. They were continually heated for another few seconds via the start and stop method observing the colour change. These AuNcgs were washed and harvested via centrifugation for further analysis. AgNcbs and AuNcgs were characterised using UV-visible spectroscopy (Beckman coulter, Brea, CA, USA), DLS (Malvern Zetasizer, ZS90, Worcestershire, UK), HRTEM (JEOL JEM1400-Plau (120 kV, LaB6) equipped with a Gatan OneView 4k camera; Tokyo, Japan), SEM (Carl Zeiss SIGMA Field Emission Scanning Electron Microscope FEG-SEM; Oberkochen, Germany), XPS (ThermoFisher ESCALAB 250Xi X-ray Photoelectron Spectrometer, Waltham, MA, USA), and Zetasizer (Malvern Zetasizer, ZS90; Worcestershire, UK).

### 2.3. Synthesis of MR-5FU Prodrug and Functionalisation of AuNcgs with MR-5FU

The extremophilic polysaccharide, MR, was chemically modified and bound with 5FU using the method adapted from [44]. MR consists of 4 constituent sugar monomers (Glu–Gal–Man–GluAcid) that repeat to form a polymer of 4.8 × 10^6^ kDa [38,39]. The reaction involved the functionalisation of MR with the chloracetyl group under chloroacetic acid and acetic anhydride treatment at 70 °C for 3 h. This was precipitated out using ice-cold water and washed several times with water and ethanol in sequence to obtain chloroacetyl-MR. After lyophilisation, a dry powder of chloroacetyl-MR was dissolved in DMSO and treated with triethylamine and 5FU, under overnight stirring at 60 °C. AuNcgs were added at the end of incubation to obtain AuNcgs coated with the prodrug MR-5FU. AuNcg-MR-5FU particles were resuspended in DMSO and washed several times using an ethanol/ether mixture (1:1, *v*/*v*). The final product was dispersed in water after ultrasonication and stored at 4 °C for further analysis and characterisation. AuNcg-MR-5FU particles were characterised using UV-visible spectroscopy, DLS, TEM, SEM, XPS, and Zetasizer.

### 2.4. Functionalisation of AuNcg-MR- 5FU Nanoparticles with Anti-CD133 Antibodies

AuNcgs-MR-5FU nanoparticles were functionalised with anti-CD133-FITC antibodies (Fisher Scientific #11-1331-82; Waltham, MA, USA) for the targeting purpose. The spontaneous binding method was adopted for tagging anti-CD133 antibodies to AuNcg-MR-5FU. A total of 1:2000 dilution of stock concentration was made before adding to the nanoparticle solution. The mixture was constantly stirred under controlled dark conditions at 4 °C, overnight. Thus, obtained TMFG nanoparticles were washed twice with PBS, to remove the unbound or loosely tagged antibodies and were used for targeting GL261cells [45,46].

### 2.5. Measurement of Au Concentration Using Microwave Plasma-Atomic Emission Spectroscopy (MP-AES)

The measurement of the Au concentration in the nanoparticles used for the biological assays and release studies was tested using Microwave Plasma-Atomic Emission Spectroscopy (MP-AES, Agilent Technologies-4100, Cheadle, UK). Five test samples of nanoparticles, along with certified reference materials (CRMs) for Au were used in this study. All samples and CRMs were prepared using the manufacturer’s standard assay procedure. Briefly, ultrapure de-ionised water (18 MΩ resistivity, Millipore, Livingston, UK), analytical grade hydrochloric acid (37% *m*/*v*, fuming) and nitric acid (69% *m*/*v*) were used in a ratio of 1:3 for the preparation of all CRMs and test samples. The final acid concentration of all solutions was 30% aqua regia. The calibration standards of 1 ppm, 10 ppm, 100 ppm, and 1000 ppm were prepared from the Au single element standard. Standards were also prepared in 30% (*v*/*v*) aqua regia. All measurements were performed using the Agilent 4110 MP-AES with a fully integrated autosampler and humidifier accessory. The instrument was fitted with a peristaltic pump to allow a modified pump tubing configuration for faster sample uptake. The sample introduction system consisted of a nebuliser, a single-pass glass cyclonic spray chamber, and an easy-fit torch. Read time was decreased to 1 s for the maximum sample throughput and the ideal nebuliser flow was easily determined using the Optimize Nebulizer Flow tool in the MP Expert software 10.1. All other method parameters used default settings. Measurements were made in triplicates from each sample and the average was plotted.

### 2.6. In Vitro Drug Release Using Slide-A-Dialysis Cassette Method

A known concentration of 5FU was taken initially and serially diluted for the preparation of the calibration curve. All individual dilutions were measured under a UV plate reader (Molecular Devices Spectra Max 340PC, Hampton, NH, USA) at 260 nm or with the HPLC-UV/LCMS method for obtaining a standard curve for 5FU. The coefficient of regression was checked, and a standard curve can be used for analysing unknown concentrations of 5FU. Before starting the qualitative in vitro drug release studies, the total amount of drug encapsulated within the nanoparticles was calculated. After washing the drug-encapsulated nanoparticles, 0.5 mL of spent wash solution was collected and the unbound free-drug concentration was identified. Later the encapsulation efficiency was back-calculated using the unbound 5FU concentration obtained from the supernatant. The supernatant collected was subjected to concentration analysis using a UV-plate reader at 260 nm and HPLC-UV/LCMS. The encapsulation efficiency was obtained from the standard curve for the plotted 5FU. In vitro, drug release studies were performed using physical (disruption by ultracentrifugation) and chemical methods (pH-based). Initially, physical methods using ultrasonication were completed for 2 h at 37 °C and the solution was spun down to collect the supernatant for 5FU analysis using LCMS, followed by a pH-based method. This method used 4 different drug release mediums: (i) phosphate-buffered saline (PBS, pH 7.4), (ii) phosphate-buffered saline (PBS, pH 3.15), (iii) acetate buffered saline (ABS, pH 3.76) and normal saline. An incubator was set at 37 °C with four different magnetic stirrers set with the above-mentioned release mediums. Slide-A-dialysis cassettes (MWCO 10 kDa) were injected with 0.5 mL of TMFG nanoparticle solution and placed in the release medium, respectively. The set-up was constantly stirred at 100 rpm speed for 14 days. The samples were collected from the respective release mediums to check the release of the free drug 5FU under UV-vis spectroscopy. An equal volume of fresh dissolution medium was replaced to maintain the sink conditions after drawing the sample for measurement each time and they continued being stirred. This process continued for various periods starting from 0 min, 15 min, 30 min, 1 h, 2 h, 4 h, 8 h, 12 h, 24 h and continued until 14 days. UV-Vis spectra were measured and captured.

### 2.7. Tumour Homogenate Assay for Ex Vivo Drug Release Study

The determination of 5FU released from TMFG was studied using the ex vivo method using glioma tumour homogenate. All work was conducted in compliance with a UK Home Office licence, which is regulated by the Animals (Scientific Procedures Act 1986) Act 1986. C57BL/6 mice were subcutaneously injected with GL261 cells. It was evident that GL261 cells consistently initiated subcutaneous tumours in C57BL/6 mice on day 12 [47]. GL261 cells were maintained by thrice weekly passaging in DMEM media supplemented with 10% FBS under controlled normoxia conditions. GL261 cells were harvested and mixed 1:1 with a matrigel matrix and inoculated into the mice. The final cell concentration was around 1 × 10^6^ cells/100 μL in the matrigel. The mice were monitored for tumours daily and were maintained until the tumour size was palpable, at which point the tumour volumes were recorded. The tumour volume was obtained by measuring the short and long diameters and then expressed as a mean product of the diameter ± SE. Calliper-based measurements have limitations over image-based measurements of tumour volume, as they do not take into account the fact that the stack’s height varies while its length and width do not. Therefore, the more the stack’s height differs from its breadth, the less accurate calliper-based measurements become. However, we have completed all possible measurements and average values were considered to calculate the tumour volumes. The weights of the mice and tumour volumes were monitored twice weekly. The mice were maintained until reaching a maximum tumour burden of 12 mm mean diameter. Upon culling, tumours were visually inspected for gross levels of heterogeneity/central necrosis and stored as snap-frozen for ex vivo studies.

The concentration of 5FU in the tumour homogenate sample was determined using the following LC-MS/UV method. Analysis was completed via HPLC-MS/MS using electrospray ionisation. Mass hunter Software 10.1was used; ESI ionisation with negative ion polarity and UV-DAD at 265, 255 and 230 nm were used for detection. The analytical column used was Sphereclone (Phenomenex, Torrance, CA, USA) 5 mm ODS2 (150 × 4.6 mm); the mobile phase used was water and methanol. All samples and calibration standards for analysis were prepared in the same way. A total of 75 µL of centrifuged incubation samples/ standard solution were mixed with 75 µL acetonitrile. During the method establishment and sample analysis, samples were extracted through aliquoting (10.0 μL) into a 96-well plate followed by the addition of ISWS (20.0 μL). The plates were sealed and vortex-mixed. A total of 150 μL of acetonitrile was then added to precipitate the proteins. The plate was sealed, mixed for 3 min and centrifuged (at 3000 rpm, +4 °C, 5 min). Then, 10 μL of the resulting supernatant was then transferred to a clean 96-well plate, followed by the addition of 250 μL of 0.1% formic acid water. The plates were sealed and mixed well before analysis. The method was established using an initial batch comprised solely of calibration, QC and blank samples to assess the precision and accuracy, linearity, sensitivity, selectivity and carryover of the method. All study samples were analysed in a subsequent single batch. The required specification for results from the calibration and QC samples was ±20% (±25% at LLOQ) of the prepared concentration.

### 2.8. In Vitro Cytotoxicity Assay

The quantification of the cytotoxicity of TMFG NPs against GL261 cells was made using an MTS assay. MTS assay was performed as per the standard protocol from the manufacturer [48]. The stock culture of GL261 cells was maintained in serum-free DMEM/F12 medium supplemented with Mouse-EGF and basic FGF growth factors under 5% CO_2_ at 37 °C in a humidified atmosphere. The used media was replenished with fresh media every 2 days. The cells were grown until a confluent growth was observed under the microscope. Once the plate attained cellular confluency, the cells were washed with fresh DMEM/F12 twice and trypsinised for 2 min by adding 1–2 mL of 0.25% trypsin-EDTA. After 2 min, the trypsin-EDTA was neutralised by adding the fresh DMEM/F12 medium into it. The cells were then spun down at 3000 rpm for 3 min and cell pellets were collected. A cell count was performed via the trypan blue exclusion method. GL261 cells were diluted and seeded to achieve a final concentration of ~5000 cells per well in a 96-well plate. All plates were incubated for 24 h at 37 °C under the same controlled conditions. After 24 h, test samples were added in a fresh medium at appropriate concentrations to the test wells leaving the negative control with media alone. Plates were treated with varying concentrations of TMFG nanoparticles (0.0001, 0.001, 0.01, 0.1, 1 mg/mL) and their anticancer activity was compared with equal concentrations of free drug 5FU and TMFG-NT NPs, respectively. All plates were incubated for 24–72 h with test samples under the same conditions. At the end of incubation, 20μL of the MTS reagent (CellTiter 96 reagent, ThermoFisher, Waltham, MA, USA) was added and incubated at 37 °C for 1–4 h in a humidified CO_2_ atmosphere. The absorbance was recorded at 490 nm using a 96-well plate reader. The corrected absorbance at 490 nm (Y-axis) versus the concentration of growth factor (X-axis) was plotted. The following formula was used for identifying the % of cytotoxicity:% Cytotoxicity = (Control − Test) × 100/Control.

### 2.9. ROS Measurement Using 2′,7′-Dichlorofluorescein Diacetate (DCFDA) Assay

DCFDA assay is a method used to measure hydroxyl, peroxyl and other reactive oxygen species (ROS) activity within the cell using the fluorogenic permeant reagent DCFDA. GL261 cells were grown until confluency was obtained and test agents (5FU, TMFG-NT and TMFG) were treated for 1–6 h. The assay protocol is based on the diffusion of DCFDA into the cell. It is then deacetylated using cellular esterases to a non-fluorescent compound, which is later oxidised using ROS into 2′,7′-dichlorofluorescein (DCF). DCF is highly fluorescent and is detected using fluorescence with excitation/emission at 485 nm/535 nm using a plate reader (Molecular Devices Spectra Max 340PC). Here, we have analysed TMFG NPs and compared them with TMFG-NT and 5FU for the production of ROS; the graph was plotted against the concentration for time intervals at 1 h, 2 h, 4 h and 6 h, respectively.

### 2.10. Cell Binding and Fluorescence Microscopy

The binding specificity of the anti-CD133 antibody functionalised TMFG NPs towards GL261 cells was tested using fluorescence microscopy. TMFG-NT and TMFG NPs were added to GL261 cells. After 24 and 48 h of incubation, unbound particles were washed twice and stained with DAPI. Finally, cells were washed and viewed under a fluorescence microscope (Evos XL Core, Seoul, Korea) for green fluorescence to demonstrate the specific binding of TMFG NPs to GL261 cells.

### 2.11. Generic Caspase Assay

GL261 cells were grown until confluency was obtained and test agents (5FU, AuNcgs, TMFG-NT and TMFG) were treated for the activation of the caspase enzyme, which is a widely accepted indicator of cell apoptosis. The green fluorometric dye irreversibly binds and retains with the apoptotic cells when they are exposed to caspase enzymes. This was measured using a flow cytometer (ThermoFisher, Waltham, MA, USA) at 480/520 nm (Ex/Em).

### 2.12. Live/Dead Assay

Live/dead assay was performed using calcein-AM and propidium iodide (PI) staining and flow cytometric analysis. The viability of the cells was monitored based on the green fluorescence shown via the cells with intact cell membranes. Cell membrane disruption can cause PI absorption and the staining of the genetic material, which is measured in red fluorescence using a flow cytometer. GL261 cells were treated with 1 mg/mL of test agents (TMFG, TMFG-NT, 5FU and AuNcgs) for 24 h on a 12-well plate and washed at the end of incubation using a PBS solution. Cells were then treated with calcien-AM and PI stains as mentioned earlier, and fluorescence was measured in the FL1 (green) and FL3 (red) channels using a BD accuri flow cytometer (ThermoFisher, Waltham, MA, USA).

## 3. Results and Discussion

### 3.1. Synthesis of MR-5FU and TMFG Nanocages

AuNcgs were prepared using the microwave-assisted synthesis method as mentioned elsewhere (Figure 1) [43]. AuNcgs obtained were washed thoroughly before the functionalisation steps. Based on the previously published method for modifying 5FU with a galactose-containing polymer like pectin [44], we modified galactose and galacturonic acid-containing MR for the present study. Chemically modified MR, when derivatised with 5FU, yielded a new stable prodrug MR-5FU. Here, we test the function of a similar carbohydrate prodrug, MR-5FU against glioma, GL261. The chloracetyl group was introduced into the galactose units available in the monomers after treatment with chloroacetic acid and acetic anhydride. Thus chloracetyl-MR was subject to derivatisation with 5FU to obtain the prodrug MR-5FU. The chemical synthesis of the MR-5FU prodrug and functionalising on the surface of AuNcgs was performed for the first time. This method was finally adapted and optimised for the synthesis of TMFG nanoformulation. The non-targeted version of TMFG (TMFG-NT) was then subjected to functionalisation with fluorescein isothiocyanate (FITC)-labelled CD-133 monoclonal antibodies. The evaluation of the release behaviour of 5FU from MR-5FU was performed after functionalising onto the surface of AuNcgs. TMFG nanocages were synthesised with a central core of AuNcgs produced through a microwave oven-assisted method [3,43]. The steps involved in the synthesis were briefed in the schematic representation (Figure 1).

### 3.2. Characterisation of TMFG Nanocages

The characterisation of AuNcgs was performed using high-resolution transmission electron microscopy (HRTEM) (JEOL JEM1400-Plau (120 kV, LaB6) equipped with a Gatan OneView 4k camera; Tokyo, Japan), (Figure 2A). The TEM image of AuNcgs shows the hollow interior and opening in the corners and faces of the nanocages. Whereas the TMFG nanocages (Figure 2B) showed a hazy pattern on the surface of the nanocages showing the outer layer of subsequent functional moieties, MR-5FU and the Anti-CD133 Abs. In the inset, the MR-5FU coating can be seen with a thick outer layer surrounding the nanocage shell. The qualitative assessment of the 5FU chemical modification onto MR and its subsequent nanocage attachment was completed using UV-visible spectrophotometry (Beckman coulter, Brea, CA, USA) obtaining UV-vis spectra of AgNcbs, AuNcgs, free-5FU, TMFG-non-targeted (TMFG-NT) and TMFG (Figure 2C). The absorbance at 425 nm of AgNcb shifted to 800 nm as the galvanisation process resulted in AuNcgs. The free drug 5FU has an absorption maximum at 230 nm and 266 nm. TMFG-NT showed a prominent 5FU peak, however, the thick coating of MR-5FU over the surface of nanocages masked the absorbance of AuNcgs at 800 nm. The successful anti-CD133 MAb functionalisation on the surface of TMFG-NT was demonstrated by the addition of a strong peak of FITC together with the drug spectra. This was generated and shown as a shift in peak towards 400 nm. It was evident that the absorption spectra of the TMFG have an obvious broad peak in the range of 266 nm–460 nm, attributed to the drug and FITC present in the final nanoparticle formulation [49,50,51]. The DLS measurement of AuNcgs, TMFG-NT and TMFG showed a gradual variation in the size of the nanocages before and after the series of functionalisation processes (Figure 2D–F). The DLS diagram of AuNcgs (Figure 2D) showed a maximum size distribution range of 70 ± 20 nm, which on subsequent functionalisation with MR-5FU increased to 276.5 ± 137 nm (Figure 2E). However, the final DLS size distribution of the TMFG nanoparticles was measured as 324.5 ± 10 nm (Figure 2F). The variation in DLS size distribution for AuNcgs and TMFG-NT can be attributed to the swelling of polysaccharides on hydration. It is a concern that when the hydrodynamic diameter (HD) is larger, there is less cell penetration and bioavailability under in vivo conditions. However, the gold nanoparticle used here has a size distribution of 70 ± 20 nm, which on polymer passivation increased to 324.5 ± 10 nm on swelling. This can be possibly controlled by reducing the wt% of the polysaccharide chloroethyl-MR to reduce the HD. The SEM characterisation data of TMFG nanocages showed excellent consistency with TEM and DLS information collected (Figure 3). The differences in the size distribution of nanocages before and after functionalisation were well-defined from SEM characterisations. ζ-potential measurements showed an increase in negative potential from −24.4 ± 0.34 mV to −15.7 ± 1.11 mV from TMFG-NT to TMFG, respectively. This indicates that the TMFG formulation synthesised is a stable suspension in nature, however, the stability comparatively decreased upon antibody functionalisation.

**Figure 2 biomedicines-12-00934-f002:**
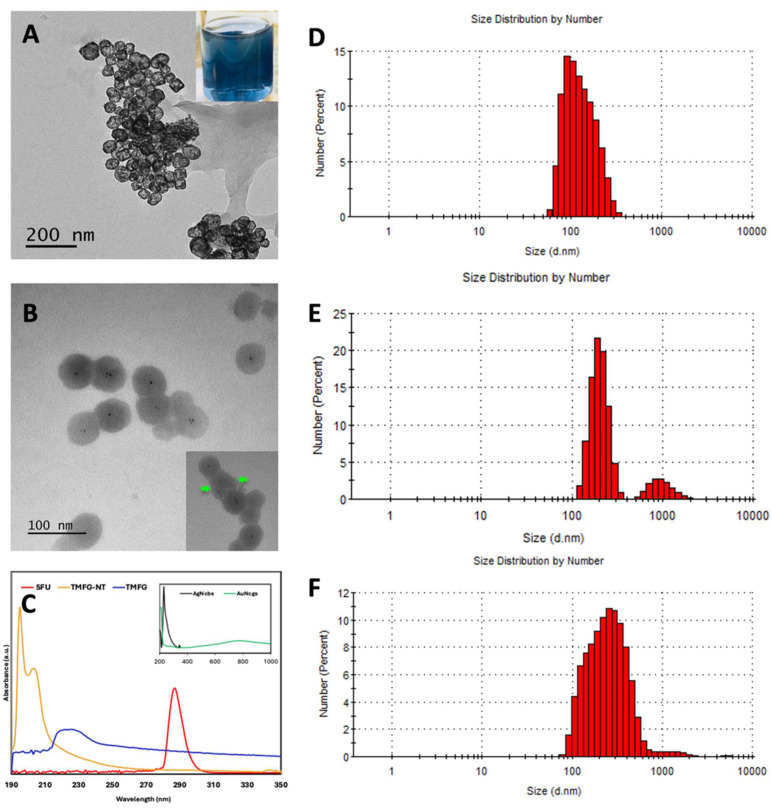
(**A**), TEM image of hollow gold nanocages (AuNcgs), with the photographic image of the hollow AuNcg solution (inset); (**B**), TEM image of TMFG nanoparticles, black dots in the corners are the crystal nucleation points of the nanocage structure during the self-assembly process with magnified image of TMFG showing the layer of MR-5FU covering (inset—green arrow); (**C**), UV-Visible spectroscopic data for silver nanocubes (AgNcbs) and AuNcgs (inset), 5-fluorouracil (5FU), TMFG-NT and TMFG nanoparticles; (**D**), DLS size distribution graph of AuNcgs; (**E**), DLS size distribution graph of TMFG-NT; (**F**), DLS size distribution graph of TMFG nanoparticles.

**Figure 3 biomedicines-12-00934-f003:**
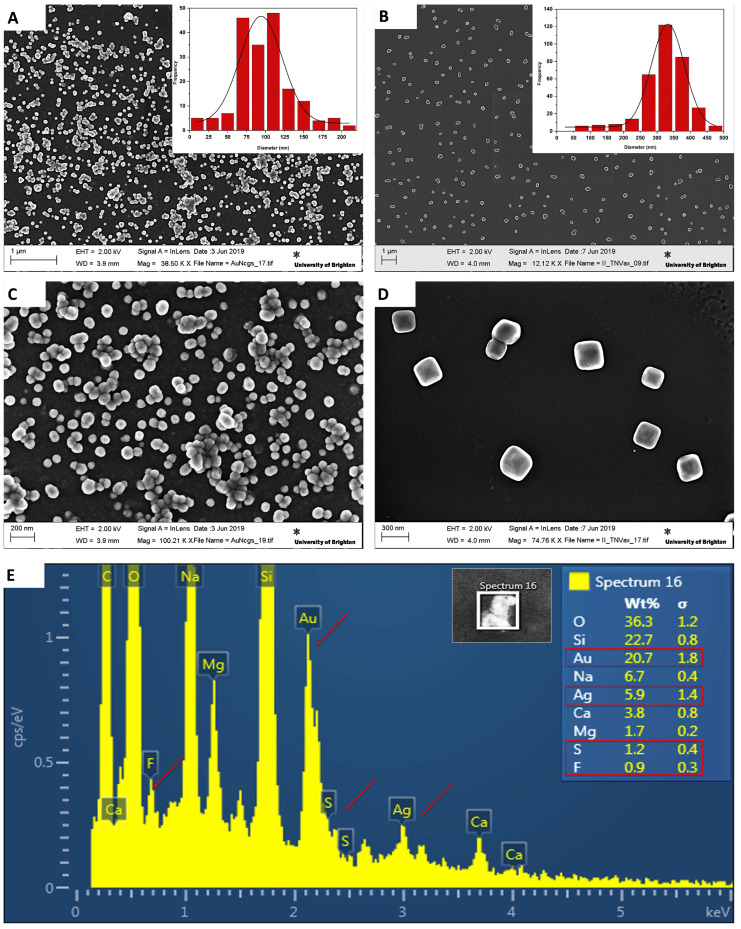
(**A**), SEM image of hollow gold nanocages (AuNcgs) and its corresponding size distribution graph (inset); (**B**), SEM image of TMFG nanoparticles and its corresponding size distribution graph (inset); (**C**)**,** magnified SEM image of hollow AuNcgs at higher resolution; (**D**), magnified SEM image of TMFG nanoparticles at higher resolution; (**E**), SEM-EDS spectrum for TMFG nanoparticles, showing its constituent elements and the signature elements like Ag, Au, S, F (red arrow); EDS scan area and weight percent (W%) of the constituent elements are shown in the inset. The surface chemistry of the TMFG nanocage is very important to understand the chemical bond disposition, especially with 5FU, MR and AuNcgs. For TMFG nanocages, the XPS spectrum measured the prominent peaks for C 1s, O 1s, N 1s, F 1s, Au 4f, and S 2p. Survey scan spectra for C 1s were measured and compared for all the individual components of TMFG, viz., AuNcgs, 5FU, MR, MR-5FU, anti-CD-133 Ab and TMFG (See Appendix A). It was observed that the signature elements increased in their weight percentage (w%) as a result of sputtering. The w% of Au, F, and S increased from 0, 0.31, and 0.75 to 0.48, 0.58 and 0.97, respectively (Table 1). This has been attributed to the gradual etching of the protein later from MAb that reveals the MR-5FU component followed by the AuNcg central core. This can be further confirmed by the element N (from MAb), from its w% reducing from 7.17 to 4.41. The presence of MR-5FU on the surface of the gold nanosurface was well-explained by the etching data obtained at different periods (0–240 s). On successive sputtering, the out layer of the MAb started to etch out and the polymer-drug layer as well as the gold nanoparticle layer started to reveal. This was explained by the gradual enhancement of the Au 4f _7/2_ peak (Figure 4B). It shows how the gold signal increased with the sputtering of the organic material. This visualisation is supported by the survey scan data, which show a decrease in the N1s signal (Table 1, supporting the sputtering of antibody) and the appearance of the Au4f signal. F1s signal also increased with sputtering (Figure 4C–E). Before sputtering, a prominent C–F covalent bond and C–F ionic bond were shown. However, on subsequent sputtering for 240 s, the C–F ionic bond started to decline (Figure 4D,E). Thus, indicating that the 5FU molecules on the surface of the polymer are more ionically attached compared to a stronger covalent interaction towards the inner core of the TMFG nanocage complex.

**Figure 4 biomedicines-12-00934-f004:**
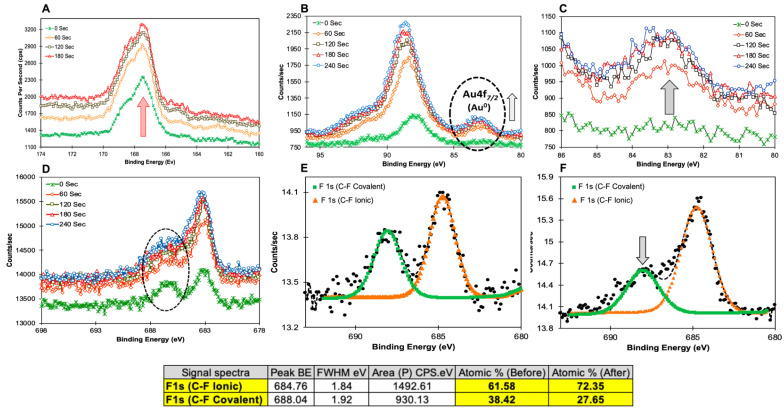
XPS measurement spectra—(**A**), High-resolution XPS spectra for S 2p peak at 167.9 eV before (0 s) and after the successive sputtering process (60 s, 120 s, 180 s). S 2p signal was significantly increased on sputtering; (**B**), high-resolution XPS spectra for Au 4f peak at 89.5 eV and 84 eV before (0 s) and after the successive sputtering process (60 s, 120 s, 180 s). Signals of both the peaks were increased however, Au 4f 7/2 peak at 84 eV was significantly increased corresponding to Au° (inset); (**C**), Au 4f 7/2 peak area was expanded to show the increase in signal during the sputtering process; (**D**), high-resolution XPS spectra for F 1s peak at 89.5 eV and 84 eV before (0 s) and after the successive sputtering process (60 s, 120 s, 180 s). (**E**,**F**), deconvoluted peaks of F 1s showing the C–F (covalent) and C–F (ionic) bonds and corresponding atomic % before (0 s) and after sputtering (180 s). It was observed that the C–F ionic bond strength increased compared to a decrease in the covalent bond, showing the strong interaction of drug 5FU with Mauran polysaccharide, which makes the difficulty in the drug release from the TMFG nanoparticles.

EDS measurements showed the presence of signature elements Au, Ag, S and F from the TMFG nanocages accordingly (Figure 3E). Successful drug and MAb binding was confirmed with the presence of F and N signals. The interaction of these elements and their surface composition along with chemical bonds formed during the nanocomplex formations is well-explained with XPS spectroscopy.

### 3.3. Encapsulation Efficiency TMFG Nanocages In Vitro and Ex Vivo

The evaluation of the release behaviour of 5FU from MR-5FU was performed after functionalising onto the surface of AuNcgs. The encapsulation efficiency of 5FU in the TMFG nanoparticle solution was found to be 75%, measured using the detection of unbound 5FU in the supernatant of wash solutions. The drug release in four different buffer mediums, phosphate-buffered saline (PBS, pH 7.4), (ii) phosphate-buffered saline (PBS, pH 3.15), (iii) acetate-buffered saline (ABS, pH 3.76) and normal saline (pH 7.0), were performed for 13 days under controlled conditions at 37 °C with constant stirring. Two strong peaks were detected at 230 nm and 266 nm. The strong signature peak of 5FU at 266 nm corresponds to the constituent broken molecule from the MR-5FU complex under the ABS medium. The drug release peak started to appear on the thirteenth day and the peak completely subsided on the next day. The broad UV-Vis peak obtained (See Appendix A) at 266 nm could be due to the strong absorption shown in the range of 5FU by the small drug-conjugated molecule released with the drug 5FU in it. This was confirmed using the LCMS analysis, which did not show any peak for the free drug 5FU. Thus, it confirms that the strong chemical bonds formed between MR and 5FU result in the breakage and release of small active prodrug molecules from the MR-5FU complex. The release profile of free 5FU from the formulation was very low under most of the in vitro conditions tested, however, an active molecule of 5FU released in small molecule form was detected using the UV-Vis spectrum. This shows that the formulation is either highly stable under all the in vitro test conditions studied or the release profile of 5FU is sustained for a longer period, which is unable to be found under in vitro dissolution medium methods. The release profile observed with the ex vivo method, using liver and tumour homogenate assay, suggested that the release of the free drug occurs after a certain period (Figure 5A,B). The release profile showed under the liver homogenate condition was negligible compared with its 0 min incubation to 5 min, and thereafter it detected much lower. However, there was a consistent increase shown under tumour homogenate conditions while tested (Figure 5B). This shows that the TMFG formulation has a tumour-specific release. The tumour microenvironment of the GL261 tumour favours the release of 5FU from the MR-5FU prodrug complex of the TMFG nanoformulation. The test was performed only for 5 h due to the instability of the enzymes and cofactors under ex vivo conditions.

### 3.4. Cytotoxicity of TMFG Nanocages

The anti-cancer activity of TMFG nanoparticles and other test agents was tested and confirmed via an in vitro cytotoxicity assay performed against GBM cells (Figure 5C–F). GL261 and GBM cells were incubated to obtain a 90% confluent growth before starting the MTS-cytotoxic assay. Firstly, five different concentrations (8.25, 16.5, 33, 66, and 132 µg/mL) of AuNcgs were tested for cytotoxicity (Figure 5C) and 132 µg/mL concentration of AuNcgs was optimised for TMFG preparation. The concentration of the gold nanoparticles was tested and analysed using the MP-AES for five different concentrations of the AuNcgs. The optimal concentration of AuNcgs used in the cytotoxicity studies of TMFG particles was initially analysed for respective gold toxicity towards the GL261 cells and the same concentrations were used for all biological assays conducted under this study. The cytotoxicity of TMFG nanoparticles was evaluated against free drug-5FU and TMFG-NT particles. MTS assay showed a high cytotoxic effect of TMFG nanoparticles on GL261 cells similar to free drugs. Figure 5D–F show the combined MTS assay results for 5FU, TMFG-NT and TMFG against GL261 cells for 24 h, 48 h and 72 h, respectively. Five different concentrations (0.0001 mg/mL, 0.001 mg/mL, 0.01 mg/mL, 0.1 mg/mL, and 1 mg/mL) of test formulations were used and compared against the positive control (5FU) and negative control (cell culture medium). Overall, TMFG-NT and 5FU showed a gradual decline in cell viability compared to a sudden drop in TMFG at 72 h. Both 5FU and TMFG-NT have some similarities in their toxic actions. The anticancer effect shown by TMFG-NT and 5FU was gradual in a time-dependent manner; around 40% of cells were killed after 48 h. However, there was no significant anti-cancer effect showed by TMFG in the 24 h or 48 h incubation period. However, there was a significant decline in the cell viability after 72 h incubation with TMFG. Almost 80% of cancer cells were killed using TMFG. Here, the cytotoxic effect shown by TMFG is almost 10% higher than its non-targeted version and equally comparable with its positive control-free 5FU. The IC-50 values shown for all TMFG and TMFG-NT agents were highly comparable with free drug 5FU at 72 h study with a concentration of 0.0001 mg/mL. This shows the excellent potency of the test formulation against CD133+ targeted therapy. The increased activity after 72 h can be attributed to the specific target interaction and uptake by the CD-133+ cancer cells and receptor-mediated uptake by GL261 cells. It has been proved that passive targeting relies on an enhanced retention and permeation effect, whereas targeted delivery relies on receptor-ligand interactions that improve selective accumulation to target sites [52]. Thus, showing the TMFG nanoformulation is equally good to the free drug 5FU with a sustained effect and target-specific delivery of the payload.

### 3.5. ROS Generation via the Action of TMFG Nanocages

ROS generation via the action of TMFG NPs was measured in vitro using DCFDA assay. The production of hydroxyl, peroxyl and other ROS molecules can be indirectly measured using the green fluorescence produced by the DCFDA reagent. The microplate analysis revealed the gradual rise in ROS production after treatment with test agents, viz. 5FU, TMFG-NT and TMFG. Here, we keep the negative control zero for ROS generation, as it includes only untreated cells growing in the medium. It was seen that the amount of ROS released was high at the end of the sixth hour of incubation in all three test compounds analysed (Figure 5G–I). However, the rate of increase shown in the positive control free drug 5FU was significantly high due to its acute toxicity and quick absorption of the drug molecule by cancer cells (Figure 5G). 5FU has a half-life and circulation time of 15–20 min in the host body. However, the ROS generation from TMFG-NT (Figure 5H) and TMFG (Figure 5I) treated cells gradually increased in a sustained manner, compared to 5FU. Among the targeted and non-targeted NPs, TMFG showed a slower release of ROS compared to TNFG-NT. This can be attributed to the cell-specific interactions of TMFG particles to the receptors and its subsequent endocytosis process. The slow release and accumulation of ROS on the specific binding of TMFG will increase the dwelling time and can enhance the induction apoptosis in cancer cells through oxidative damage and associated cell death mechanisms more sustainably. Similar therapeutic nanocage-mediated multiple cell death mechanisms in breast cancer cells were reported in our previous work [3,53]. Hereby, it was shown that the TMFG NPs have a time-dependent ROS generation based on the concentration of NPs acted upon the GL261 cells.

### 3.6. TMFG Nanocages Selectivity towards CD-133 Binding on GBM Cell Line and Caspase Assay

TMFG nanoparticles were tested and evaluated for selectivity towards CD-133 binding on the GBM cell line and GL261 using fluorescence microscopy (Evos XL Core). Fluorescently tagged anti-CD133 antibodies on the surface of TMFG nanoparticles showed green fluorescence on binding to CD133+GL261 cells (Figure 6A–P). GL261 cells were treated with TMFG and TMFG-NT nanoparticles for 24–48 h, respectively. The TMFG showed uniform green fluorescence on the cell surface whereas the TMFG-NT did not. It is evident from the images that the TMFG cells undergo receptor-mediated endocytosis compared to the non-specific interaction of the TMFG-NT with cells. The former has a greater potential for binding and specific ingestion via the cancer cells compared to the non-specific interactions. The fate of GL261 cells after cellular uptake was studied using a generic caspase assay (Figure 7). A FACS analysis was performed to identify the apoptosis and cell death progression on TMFG action. A generic caspase activity assay was performed to identify the cellular apoptosis after TMFG treatment for 48 h. The activation of caspase enzymes is a reliable indicator for cellular apoptosis taking place inside TMFG-wounded cells. In this assay, the activated caspase enzymes bind to the fluorescent moiety and were detected using a flow cytometer at Ex/Em = 488/520 nm. Figure 7A–D shows the caspase assay results for TMFG NPs and their comparison against their respective test agents and negative control. It was evident that the TMFG-NT and TMFG NPs induced caspase production, which is an indicator of cellular apoptosis. This has been compared with the positive control. The shift in florescence was time-dependent and the uptake of NPs increases over time as shown in the cytotoxicity results. Therefore, the apoptosis must be relative to the sustained release pattern of the active drug complex from TMFG NPs.

### 3.7. Live/Dead Assay

A live/dead assay was conducted using propidium iodide (PI) and calcein AM (CalAM) staining for the identification of cells undergoing death mechanisms after the treatment with TMFG NPs for 24 h. Cell viability and cytotoxicity were measured using green and red fluorescence, respectively, using flow cytometry. Cell viability is measured based on the intactness of the plasma membrane in the live cells and their intracellular esterase activity. Compromised cell membranes lead to the permeation of PI, thereby staining nucleic acid and resulting in red fluorescence. The action of TMFG NPs was compared with their counterparts. Apoptosis and necrosis are the two major pathways that lead to cell death. Results obtained from the generic caspase assay are in line with the observations of the live/dead assay (Figure 7E–I). It was observed that the TMFG NPs showed 22.05% (Figure 7I) of cell death compared to the NC of 14.0% (Figure 7E), whereas TMFG-NT and 5FU showed 26% (Figure 7H) and 24.8% (Figure 7G), respectively. The cell populations were gated based on their fluorescence uptake as live, early apoptotic and late apoptotic cells. The death percentage was calculated as a sum of early and late apoptotic cell populations observed. The slight increase in the TMFG-NT can be attributed to its non-specific interaction and uptake through the endocytosis process.

## 4. Conclusions

We have shown how to use AuNcgs to precisely target CD133+ glioblastoma cells while nanotargeting MR-5FU, a novel bacterial polysaccharide prodrug that was created from a low bioavailable drug like 5FU. According to in vitro cytotoxicity tests, GL261 cancer cells may be effectively killed using TMFG nanoparticles containing the highly sulphated extremophilic bacterial polysaccharide MR-based 5FU prodrug, chloracetyl-MR-5FU. We have achieved a 75% encapsulation efficiency for 5FU during the prodrug synthesis on the surface of MR. The release profile of 5FU from the formulation was very low under most of the in vitro conditions tested. This implies that the formulation is extremely stable under every in vitro test condition that has been conducted. However, the ex vivo tumour homogenate assay’s release profile revealed that the free drug was released after a specific amount of time during the incubation period. Because the MR and chloracetyl–MR-5FU complex are a unique polysaccharide and a novel combination prodrug, respectively, the structure and release of the prodrug’s active component must be determined. Before moving on to in vivo efficacy, we need strong PK data under a stable in vivo system because this is a unique drug combination that is nano-encapsulated. The 5FU release strategy observed during the in vitro and ex vivo assay suggested its successful therapeutic effect and future possible action of MR-5FU under in vivo conditions, which is yet to be tested and identified using appropriate glioma murine models. Also, the targeted binding to CD133+ glioma cells suggests the specific interaction and binding to putative BTICs that might contribute to stemness in the heterogenous glioma population. Thus, our results indicate that the novel strategy of developing bacterial sulphated polysaccharide prodrugs with less bioavailable cancer drugs can be beneficial for bringing increased cytotoxicity to glioma cells via nanotargeting, thereby bringing more combination therapeutic strategies for cancer treatment to light.

## Figures and Tables

**Figure 1 biomedicines-12-00934-f001:**
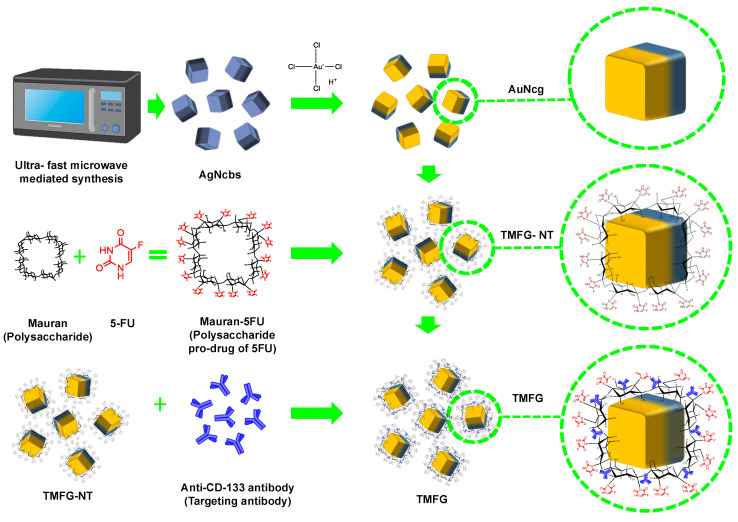
Schematic representation of the synthesis of Therapeutic Mauran–Fluorouracil Gold (TMFG) nanocages.

**Figure 5 biomedicines-12-00934-f005:**
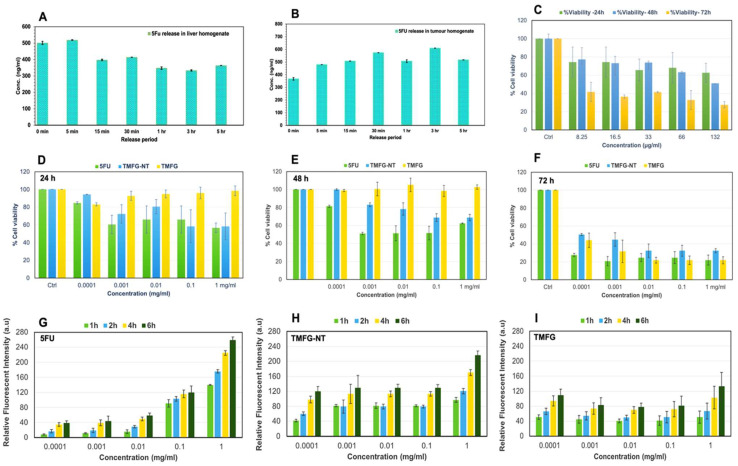
(**A**,**B**) Ex vivo study conducted for 5FU release from TMFG nanoparticles using liver homogenate (**A**) and tumour homogenate (**B**); MTS cytotoxicity assay of GL261 cells for AuNcgs (**C**); (**D**–**F**) MTS cytotoxicity assay of GL261 cells under 24 h, 48 h and 72 h study (5FU, TMFG-NT and TMFG nanoparticles). Test concentrations were 0.0001, 0.001, 0.01, 0.1 and 1 mg/mL, respectively. (**D**), 24 h cytotoxicity assay data; (**E**)**,** 48 h cytotoxicity assay data; (**F**)**,** 72 h cytotoxicity assay data; (**G**–**I**) DCFDA assay for measuring the production of ROS by GL261 cells for 5FU, TMFG-NT and TMFG nanoparticles. Test concentrations were 0.0001, 0.001, 0.01, 0.1 and 1 mg/mL, respectively. (**G**), ROS assay results for 5FU (1 h, 2 h, 4 h and 6 h); (**H**), ROS assay results for TMFG-NT (1 h, 2 h, 4 h and 6 h); (**I**), ROS assay results for TMFG (1 h, 2 h, 4 h and 6 h).

**Figure 6 biomedicines-12-00934-f006:**
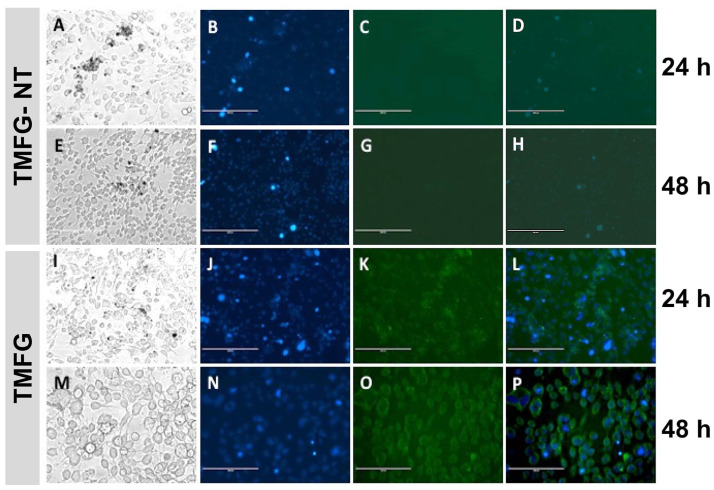
(**A**–**P**) Fluorescent microscopy for showing the CD133-based specific binding of TMFG-non-targeted (NT) and TMFG nanoparticles to GL261 cells, respectively. (**A**–**H**), Images showing GL261 cells treated with TMFG-NT nanoparticles for 24 h (**A**–**D**) and 48 h (**E**–**H**); all images are in the order of bright field, DAPI, green fluorescence from anti-CD133 Ab and merged channels; (**I**–**P**)**,** images showing GL261 cells treated with TMFG nanoparticles for 24 h (**I**–**L**) and 48 h (**M**–**P**); all images are in the order of bright field, DAPI, green fluorescence from anti-CD133 Ab and merged channels.

**Figure 7 biomedicines-12-00934-f007:**
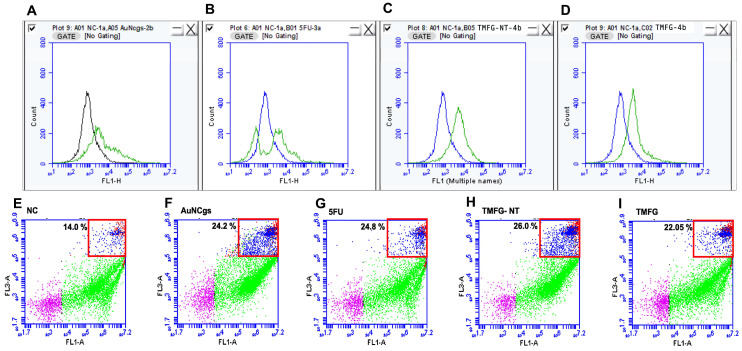
(**A**–**D**), 24 h generic caspase assay of GL261 cells for AuNcgs, 5FU, TMFG-NT and TMFG nanoparticles, respectively. All test samples were compared with negative control (NC) in all the plots. (**A**), NC and AuNcgs; (**B**), NC and 5FU; (**C**), NC and TMFG-NT; (**D**), NC and TMFG nanoparticles; (**E**–**I**), 24 h live/dead assay results for 24 h; (**E**), NC; (**F**), AuNcgs; (**G**), 5FU; (**H**), TMFG-NT; (**I**), TMFG nanoparticles; live cells (green), early apoptotic (blue), late apoptotic (red) and debris (purple). The total dead cell population was calculated with the sum of the red and blue populations gated together. FL-1A channel for green fluorescence from calcein-AM and FL3A channel for red fluorescence from propidium iodide.

**Table 1 biomedicines-12-00934-t001:** XPS quantification (atom %) of constituent elements in TMFG nanoparticles before (0 s) and after (180 s) of sputtering. The yellow portions are highlighted to show the signature elements.

Elements	No Sputtering	After 180 s Sputtering
F1s	0.31	0.58
O1s	32.67	39.58
N1s	7.17	4.41
C1s	51.95	33.72
Mg1s	0.35	2.24
Na1s	3.85	8.74
Ca2s	1.1	1.56
Cl2p	0.71	1.03
Au4f7	0	0.48
Si2p	1.14	6.69
S2p	0.75	0.97

## Data Availability

The original contributions presented in the study are included in the article/Appendix A, further inquiries can be directed to the corresponding author.

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
