# Peer review of "Combinatorial Therapy: Targeting CD133+ Glioma Stem-like Cells with a Polysaccharide–Prodrug Complex Functionalised Gold Nanocages"

_biomedicines, 2024, doi:10.3390/biomedicines12050934_

Round 1

Reviewer 1 Report

Comments and Suggestions for Authors

In the abstract, the authors assert that cancer therapies have made significant progress in activating or modulating the body’s immune response against cancers, aiming for a long-lasting anticancer effect following several significant chemoimmuno irradiations. However, they have not yet evaluated this crucial point.

The authors claim active targeting mediated by CD133, but they have not confirmed it. Please conduct an ICP analysis (gold) to evaluate this process.

Since the authors have anti-CD133 just electrostatically linked, they should describe the limitations of this technique. Antibodies require proper orientation to exert their biological effect.

The reviewer recommends that, for future studies, in order to co-localize 5-FU and gold in vivo, authors should conduct an ICP analysis. Following the 3R's rules, authors cannot repeat the experiment, but they can perform co-localization studies in cell culture.

Authors should comment on the limitations of measuring tumor volume with a caliper.

According to ISO 10993 guidelines, a 24-hour period is typically sufficient to evaluate cytotoxicity in cell culture. To assess cytotoxicity over a longer duration, authors should consider seeding cells at a lower density to prevent overgrowth effects.

Please split Figure 6. It is impossible to see the images properly.

What about hydrodynamic diameters larger than 100 nm for biomedical applications?

Figure 1C. Signal of 5FU is saturated (useless)

Section 3.3. Please provide the UV-Vis spectra in the ESI

In the in vivo experiment, what about the control with the free 5-FU?

Section 3.4 does not demonstrate any receptor-mediated interaction. Please review the previous comment

Comments on the Quality of English Language

This work needs further experiment and a deeper discussion for publication. 

Author Response

We would like to thank the reviewer wholeheartedly, for the time taken to review our manuscript and for providing recommendations to improve the work presented. We have addressed all the comments suggested and included the changes in the revised manuscript for publication. We sincerely hope that our responses will be kindly accepted by the respectful reviewer for the publication of the revised manuscript.

Many thanks.

Reviewer 2 Report

Comments and Suggestions for Authors

This work present the use of a novel strategy to encapsulate a combined polysaccharide-5FU compound and an anti-CD-133 antibody into AgNcbs nanocages to improve the delivery of the active compound in  cancer cell areas. the materials were full characterized with spectroscopic techniques and a complete battery os biological tests were performed in order to study the anticancer efficacy.

The encapsulation improves the efficiency of liberation at in vivo and ex vivo conditions of the proposed drugs, reaching a promising treatment to a wide variety of cancers. The proposed nanomaterial also is very stable at in vitro conditions.

Although the use of nanomaterials as delivery vehicles is a very explored area, the use of antibody as receptors, the nanocages and the combination of polysaccharide with effective anticancer drug such as 5-FU gives valuable knowledge to the audience to explore new combinations based in these findings.

For this kind of article, the experimental and methodological section is consistent with the information that authors want to obtain from their samples. There is no need of more controls.

The conclusion section is coherent with all the data obtained with the appropiate experiments performed, as well for the characterization as for the biological efficacy. The number and quality of references is appropriate.

I recommend accept in present form.

Author Response

We wholeheartedly express our sincere gratitude towards the learner reviewer for extending the observations and comments. We are glad that our work is appreciated and well recommended for publication in the Biomedicines in its present format.

Once again our sincere thanks for your valuable time taken to review our work.

Many thanks.

Reviewer 3 Report

Comments and Suggestions for Authors

Comments:

1- The title is not comprehensive and doesn't involve the developed nanostructure/composite. Additionally, it fails to accurately represent the purpose of the study. Amend it for clarity.

2- The abstract should provide more detail and incorporate significant results for a comprehensive overview.

3- Specify the materials used; they were not mentioned in the "2. Materials and Methods" section.

4- Evaluate and report the extent of functionalization of AuNcg-MR-5FU nanoparticles with anti-CD133 antibodies. How do we confirm that conjugation has occurred?

5- In lines 141-142, include the relevant reference to support the statement.

6- Clarify the reason for using different pH levels in the drug release assay, especially at pH 3.15 and pH 3.76.

7- Provide a proper reference for the statement "MTT assay was performed as per the standard protocol."

8- Enhance the resolution and quality of Supplementary Figure 1 for better clarity.

9- Explain the rationale for using two different wavelengths to measure 5FU in the determination of encapsulation efficiency and release tests.

10- Include the IC50 values of different test agents in section 3.4.

11- Drug release profiles are not present in the manuscript or supplementary information. Include them for a more comprehensive analysis.

Author Response

We would like to thank the respectful reviewer wholeheartedly for the time taken to review our manuscript and provide valuable recommendations to improve the work presented. We have addressed all the comments and included all possible changes in the revised manuscript submitted for publication.

We sincerely hope that our revised manuscript will be accepted for publication.

Many thanks

Round 2

Reviewer 1 Report

Comments and Suggestions for Authors

The abstract needs revision. It is not easy to read.

In order to quantify the binding of the antibodies, authors must perform a TGA analysis to check for an increase in the organic layer

Te reviewer could not find the either MP-AES methodology not results.

The co-localization with ICP in cell culture is needed. 

New text added in section 2.4 is not easy to read. The results are not properly explained.

The limitations of the caliper compared to imaging techniques are not described/disucsed in the tecxt.

There is no discussion in the text about the HD of more than 300 nm and their limitation for in vivo experiments.

The reviewer agree with authors that they onlu conducted ex vivo experiments. So, what about the control with the free 5-FU in the ex vivo experiments? 

In order to demonstrate active cell targeting, authors must perform a competition assay with TMFG and LS-7 (CD133 ligand) and analyze it using ICP

Excessive self-citation have been detected.

Comments on the Quality of English Language

The English should be revised because some paragraphs are difficult to read

Author Response

We express our deep gratitude to the considerate reviewer for dedicating time to our article.

We would like to take this occasion to apprise the reviewer of the challenges we encountered in carrying out this study and how we managed to finish most of it during the COVID-19 shutdown. This investigation was carried out as part of the coveted Marie Curie financing that the University of Brighton and the pharmaceutical company Pharmidex Pharmaceutical Services, UK skilfully won for the ‘Glioma’ research between 2017 and 2019. Toyo University in Japan provided the Mauran polymer as part of a collaborative effort; however, they do not currently make it, so we will need to culture and extract the polysaccharide, which could take many months. It took nearly six years to finish this task because of COVID lockdown issues. Now, to proceed with any further studies, we must publish this and receive the ensuing money. With humility, we submit that our work has been altered and edited multiple times, before forwarding to Biomedicines. We sincerely hope that our paper will be graciously approved for publication with this second version from "Biomedicines."

The TGA and ICP analysis currently faces several financial and technical challenges. We have conducted every study that we are currently able to do to support each study that is included in this paper. Currently, we don't have the resources to conduct additional studies, but if this is published, we will receive funding to conduct follow-up investigations. Therefore, we humbly request you to kindly accept this second revision and response to your queries and recommend our manuscript for publication.

  • The abstract needs revision. It is not easy to read.

Yes, we have rewritten the abstract with a layman’s point of view for better understanding.

  • In order to quantify the binding of the antibodies, authors must perform a TGA analysis to check for an increase in the organic layer.

We understand that thermogravimetric analyses (TGA) will show qualitatively the attachment of antibodies. Other methods like isotope labelling of the antibody could also have been employed in order to show the final attachment. However, we have shown the attachment by means of XPS depth profiling (Fig. 4), XPS narrow scan C1s deconvoluted data (Supporting figure 2) and using Electron Microscopy data showing the changes in the size of nanocages following different stages of adsorption and final TMFG. In fig. SF2. F, the C1s spectra are a combination of spectra (A), (C) and (E). There have been instances where just XPS C1s data has been analysed in order to show biomolecular attachment1,2. We have successfully shown that the changes in size and XPS C1s spectra are in line with the expected size and chemical signature changes following different stages of  TMFG nanocages formation. We are unable to carry out the TGA analysis as advised by the reviewer, but we have followed accepted analytical techniques to establish the attachment mechanism.

Reply References.

(1) Immobilization of Active Antibodies at Polymer Melt Surfaces during Injection Molding, Thor Christian Hobæk, Henrik J. Pranov, and Niels B. Larsen; Polymers (Basel). 2022 Oct; 14(20): 4426.

(2) Oriented Antibody Covalent Immobilization for Label-Free Impedimetric Detection of C-Reactive Protein via Direct and Sandwich Immunoassays; Abiola Adesina and Philani Mashazi; Front Chem. 2021; 9: 587142.

  • Te reviewer could not find the either MP-AES methodology not results.

We have included the AES methodology in the method sections and in the results as well.

  • The co-localization with ICP in cell culture is needed. 

As we humbly mentioned in our first comment section, we had only the MP-AES facility and we tried to use it for our study; the method is added to the revised manuscript submitted. ICP analysis currently faces several financial and technical challenges for us. However, we will include the same in our future follow-up study, while we perform the invivo work with these particles.

  • New text added in section 2.4 is not easy to read. The results are not properly explained.

We have now revised the section and the results were paraphrased for greater clarity.

  • The limitations of the caliper compared to imaging techniques are not described/disucsed in the tecxt.

We have now included the same in the text in the revised version submitted.

  • There is no discussion in the text about the HD of more than 300 nm and their limitation for in vivo experiments.

We have now included the limitations of the HD³ 300nm in the discussion section and a possible way to reduce/ control the HD size by tuning the synthesis process based on the weight % of the capping agent.

  • The reviewer agree with authors that they onlu conducted ex vivo experiments. So, what about the control with the free 5-FU in the ex vivo experiments? 

We want to explain the method we performed for the exvivo release experiment to clarify the point raised by the respectable reviewer: The aim of conducting the exvivo tumour homogenate study was to test the action of tumour microenvironment in the dissociation of the pro-drug from the nanoparticle complex and release to the external medium. So we used TMFG nanoparticles alone for the study. Free 5FU was not used due to the following major reasons:

  • the tumour was collected from the mice after culling at a certain point according to the Ethics policies and hence there were only enough tumour samples for the test and negative control alone for the exvivo studies. Since we conducted this experiment as part of the small initial- pilot study, our experiments were conducted with a minimal number of C57BL/2 mice.
  • Secondly, mixing the free 5FU, drug concentration with tumour homogenate cannot be considered as positive control as we were not testing the kinetics of the drug but rather the dissociation and release of pro-drug from the nanoparticle complex.

Hence, we didn't include the free-5FU as the control for the exvivo study. However, free-5FU was used as the standard to compare the release of the prodrug at the test wavelengths. But we assure you that we will include a positive control (free5FU) and the test TMFG (with prodrug), when we conduct the wide invivo study planned during the next phase of the study.

  • In order to demonstrate active cell targeting, authors must perform a competition assay with TMFG and LS-7 (CD133 ligand) and analyze it using ICP.

As we mentioned in our first comment, we have financial and technical limitations to perform and include the ICP analysis in this manuscript. But we assure you that we will include this competition assay and measure ICP in our next phase of the study. Currently, we request the respectful reviewer to understand our financial limitations and approve the manuscript for publication.

  • Excessive self-citation have been detected.

We included our previous publications to support our claims due to the use of a novel polysaccharide, Mauran, which we have introduced in the field of nanomedicine application. However, based on the suggestion from the reviewer, we have managed to remove any irrelevant self-citations. 

Reviewer 3 Report

Comments and Suggestions for Authors

The comments have been applied and are acceptable

Author Response

We wholeheartedly express our sincere gratitude towards the learner reviewer for extending the observations and comments. We are glad that our work is appreciated and well recommended for publication in the Biomedicines in its present format.

Once again our sincere thanks for your valuable time taken to review our work.